# Quantitative Detection of Defects in Multi-Layer Lightweight Composite Structures Using THz-TDS Based on a U-Net-BiLSTM Network

**DOI:** 10.3390/ma17040839

**Published:** 2024-02-09

**Authors:** Dandan Zhang, Lulu Li, Jiyang Zhang, Jiaojiao Ren, Jian Gu, Lijuan Li, Baihong Jiang, Shida Zhang

**Affiliations:** 1Key Laboratory of Optoelectronic Measurement and Control and Optical Information Transmission Technology, Ministry of Education, Changchun University of Science and Technology, Changchun 130022, China; 2School of Optoelectronic Engineering, Changchun University of Science and Technology, Changchun 130022, China; 3Zhongshan Institute of Changchun University of Science and Technology, Zhongshan 528400, China; 4Institute of Aerospace Special Materials and Technology, Beijing 100074, China

**Keywords:** composite structures, defect detection, terahertz non-destructive testing, U-Net, BiLSTM

## Abstract

Multi-layer lightweight composite structures are widely used in the field of aviation and aerospace during the processes of manufacturing and use, and, as such, they inevitably produce defects, damage, and other quality problems, creating the need for timely non-destructive testing procedures and the convenient repair or replacement of quality problems related to the material. When using terahertz non-destructive testing technology to detect defects in multi-layer lightweight composite materials, due to the complexity of their structure and defect types, there are many signal characteristics of terahertz waves propagating in the structures, and there is no obvious rule behind them, resulting in a large gap between the recognition results and the actual ones. In this study, we introduced a U-Net-BiLSTM network that combines the strengths of the U-Net and BiLSTM networks. The U-Net network extracts the spatial features of THz signals, while the BiLSTM network captures their temporal features. By optimizing the network structure and various parameters, we obtained a model tailored to THz spectroscopy data. This model was subsequently employed for the identification and quantitative analysis of defects in multi-layer lightweight composite structures using THz non-destructive testing. The proposed U-Net-BiLSTM network achieved an accuracy of 99.45% in typical defect identification, with a comprehensive F1 score of 99.43%, outperforming the CNN, ResNet, U-Net, and BiLSTM networks. By leveraging defect classification and thickness recognition, this study successfully reconstructed three-dimensional THz defect images, thereby realizing quantitative defect detection.

## 1. Introduction

Multi-layer lightweight composite structures are extensively applied in the aerospace industry owing to their high-temperature resistance and oxidation prevention. Influenced by assembly processes, service environments, and time, these structures undergo changes in their physical or chemical properties, resulting in defects, such as interface delamination and debonding, and posing safety risks [1,2]. Since the size of defects in multi-layer lightweight composites varies from a few microns to a few centimeters, larger defects can be judged by the human eye, but smaller defects are difficult to find in the macro. Based on the good penetration ability of THz in most non-polar materials, the use of THz for the non-destructive testing of multi-layer lightweight composites is an effective technical method for ensuring product quality and safety [3,4,5]. THz non-destructive testing has emerged as a cutting-edge technique capable of identifying specific defects by examining diverse spectral features within the THz wavelength range [6,7,8]. Previous studies, such as those by H. Jiang et al., utilized three feature parameters, namely, the maximum absolute value, the signal power, and the envelope area, to image mid-gap defects of various shapes [9]. Alternatively, Jiang X et al. extracted delay time differences in THz wave reflection signals to detect and image corrosion in coating steel plates [10]. While these studies employed the manual determination of defect features for image analysis, multi-layer lightweight composite structures exhibit a variety of defect types, leading to unclear patterns in the shape and position changes of THz time-domain signals. This complexity makes it challenging to distinguish between defect types. Furthermore, the significant differences in the positions of characteristic peaks and troughs for various defect types pose difficulties for traditional methods in accurately locating the depth information of defects. In order to accurately search the location of the characteristic peaks and valleys, manual intervention is often required in the identification process, introducing a subjective element to the results. Additionally, because manual identification methods are unable to meet the demands of the rapidly advancing THz non-destructive testing industry, an intelligent defect recognition method is required [11].

The application of deep learning technology in THz spectral feature extraction is extensive and numerous researchers have conducted many relevant studies [12,13]. Wang et al. utilized a one-dimensional convolutional neural network (CNN) model to detect and classify internal defects in glass fiber-reinforced polymers. Their approach outperformed long short-term memory recurrent neural networks and bidirectional long short-term memory recurrent neural network models, achieving a recall rate of up to 0.97, an F1 score exceeding 0.91, and, based on the automatic defect detection and classification, a THz image showing that the locations and depths of defects can be efficiently reconstructed [14]. Zhang et al. proposed a THz wave detection method incorporating wavelet analysis and a CNN model. By employing a residual shrinking network and a support vector machine-improved CNN, they achieved optimal performance, increasing classification accuracy to 98.91%, and, based on a wavelet analysis and the CNN-RSN-SVM model, the three-dimensional imaging method of the air gap defects was carried out. The imaging results were in good correspondence with the position and thickness of the air gap in the sample [15]. Xiong et al. employed a multi-feature fusion CNN to identify defects’ waveforms. Their automated labeling network improved the labeling speed of THz waveforms tenfold compared to traditional methods, achieving a defect recognition accuracy of 99.28% and an F1 score of 99.73%, The proposed network solves the problems of the low efficiency of the defect identification method of adhesive structures and the considerable influence of subjective factors and promotes the development of THz non-destructive testing technology [16]. Currently, many scholars are exploring the application of CNNs and recurrent neural networks in THz time-domain spectroscopy (THz-TDS) recognition [17,18].

Notably, the literature reporting the application of the U-Net network, primarily used in image processing, in the THz domain is lacking. Some researchers have applied the one-dimensional U-Net network to electrocardiogram signals. Chen et al. used a one-dimensional U-Net network for periodic segmentation and feature point detection of arterial blood pressure signals, demonstrating sensitivities of 99.79% and 99.79%, positive predictive rates of 99.99% and 99.94%, and error rates of 0.23% and 0.27% for peak values and systolic peak values, respectively [19]. Peng et al. constructed ST-Res U-Net for detecting QRS complexes and R peaks, achieving F1 values of 99.81% and 91.75%, respectively. Their model and method effectively automated the classification and annotation of electrocardiogram signals, significantly improving the accuracy of diagnosing arrhythmias [20].

In contrast to the numerous characteristic peaks observed in electrocardiogram signals, the THz signal peak is relatively single, posing a greater challenge to identifying the features of THz signal characteristics. In order to achieve defect classification and a quantitative analysis of multi-layer lightweight composite materials, a U-Net BiLSTM network was constructed in this paper to identify the characteristic peaks of terahertz detection waveforms. This study introduced enhancements to the standard U-Net network by incorporating BiLSTM into U-Net’s skip connections, thus forming a U-Net-BiLSTM network. This modification enables the automatic identification of defect types and quantitative analysis of defect thickness in multi-layer lightweight composite structures by connecting the output of BiLSTM with the corresponding feature maps of the decoder. The study involved the creation of samples with delamination and debonding defects, which were then subjected to THz non-destructive testing. Each defect sample encompassed five defects of varying sizes, allowing for the manual adjustment of the defect thickness. Subsequently, THz-TDS signals were used in the training and testing of the CNN, ResNet, U-Net, and BiLSTM network. The obtained results were compared with those from the U-Net-BiLSTM network developed in this study. Furthermore, based on the network’s recognition outcomes, three-dimensional imaging maps were reconstructed to illustrate the defects.

## 2. Experiments

### 2.1. THz Detection System

This study employed a proprietary THz-TDS instrument developed in the laboratory to detect internal defects within samples of multi-layer lightweight composite structures. The operational principle of the THz-TDS system is depicted in Figure 1. The system features a time scanning range of 300 ps, a sampling interval of 0.1 ps, a scanning frequency of 140 Hz, and a 42.5% duty cycle for the rotating delay line.

The THz-TDS system is mainly composed of a femtosecond laser, a THz radiation source (transmitting photoconductive antenna), a THz detector (receiving photoconductive antenna), and a time delay device [21]. In this paper, the reflective THz-TDS system is used to detect multi-layer lightweight composite structures. The working principle is that the laser pulse generated by the femtosecond laser passes through the beam-splitting mirror to generate pump light and probe light, respectively. The pump light shines on the surface of an InAs chip in the transmitting photoconductive antenna, thus generating a THz electromagnetic wave [22]. THz lenses are used to focus the THz waves to the depth of interest of the multi-layer lightweight composite materials; the probe light passes through the reflector and collimator to the receiving photoconductive antenna, which is used to receive the THz pulse carrying the sample material. A high-speed rotary time delay device and a high-resolution encoder are used to acquire the THz signals at a high time resolution. The THz waveform information of the whole sample plane is obtained by scanning the sample material point by point by combining the THz time-domain spectroscopy system with the planar scanning platform.

### 2.2. Sample Preparation

To replicate the defect scenarios in multi-layer lightweight composite structures, we designed and crafted samples featuring delamination and debonding defects, as depicted in Figure 2. The sample dimensions measured 50 mm × 300 mm and comprised heat-proof composite material, silicone rubber, and a metal substrate. The thickness of the heat-proof composite material, the glue layer, and the metal plate was 30 mm, 0.5 mm, and 5 mm, respectively. The design diagram of the delamination defect sample is shown in Figure 2a. The delamination defect is located in the multi-layer lightweight composite structure. The design diagram of the debonding defect sample is shown in Figure 2b. The debonding defect is located between the multi-layer lightweight composite structure and the metal plate. Varied thicknesses of the delamination defects and debonding defects were achieved by adjusting the position of the metal substrate. As shown in Figure 2, the marked A–E is predetermined defect areas, and the defect areas are specified in Table 1.

### 2.3. Dataset Labeling

Figure 3 shows a schematic representation of THz wave propagation within multi-layer lightweight composite structures. At different interfaces between the mediums, reflections and transmissions occur. The reflected signal is incorporated into the THz time-domain signal based on the propagation time, while the transmitted signal continues its downward propagation. When the THz wave moves from a low-refractive-index medium to a high-refractive-index medium, a half-wave loss transpires at the interface, resulting in a distinctive trough. Conversely, as the THz wave travels from a high-refractive-index medium to a low-refractive-index medium, a characteristic peak emerges at the interface of the two layers, persisting until it encounters the metal substrate, where total reflection occurs.

The reflection THz waveforms and signal annotations in the multi-layer lightweight composite structure samples are presented in Figure 4a–c, which represent a normal defect-free waveform, a delamination defect waveform, and a debonding defect waveform, respectively. In the bonding area of the defect-free sample, the THz waveform exhibits distinct peaks. Peak 1 represents the surface reflection peak on the surface of the multi-layer lightweight composite structure; peak 2 represents the reflection peak at the interface between the multi-layer lightweight composite structure and glue; and peak 3 represents the reflection peak at the interface between the glue and metal substrate. In the bonding area of the delamination defect sample (Figure 4b), the waveform displays noticeable fluctuations, where characteristic trough 4 and feature peak 5 emerge between peaks 1 and 2, serving as distinctive indicators of the delamination defect. The B-Scan graph highlights the region with delamination defect information, enclosed by the red dashed circle. Similarly, in the bonding area of the debonding defect sample (Figure 4c), the waveform exhibits prominent fluctuations, where characteristic trough 6 and peak 7 emerge between peaks 2 and 3, acting as indicative features of a debonding defect. The B-Scan graph highlights the area with debonding defect information, enclosed by the red dashed circle. Leveraging the unique positions of the THz characteristic peaks associated with different defect types, labeling was assigned to the positions of these feature peaks in the time-domain signal. On this basis, the THz signals were divided into three categories—the invalid region, the delamination defect region, and the debonding defect region. The manual annotations are displayed below the waveforms in the figures. In this study, the defect thickness was precisely determined using a caliper. Samples with defect thicknesses of 0.2, 0.4, 0.6, and 0.8 mm were chosen to construct the THz-TDS detection waveform dataset. This dataset encompassed 2000 defect-free THz-TDS waveforms, 2000 waveforms with delamination defects, and 2000 waveforms with debonding defects. The dataset was divided into a training dataset and a validation dataset in an 8:2 ratio.

### 2.4. U-Net-BiLSTM Network Architecture

To effectively detect various defect types and thicknesses in samples of multi-layer lightweight composite structures, this study integrated BiLSTM into the U-Net network, creating a one-dimensional U-Net-BiLSTM network model. The U-Net structure, with its encoder–decoder architecture [23], captures spatial features within sequences, thereby facilitating the localization and classification of signal features. The BiLSTM, with its ability to consider both past and future context information, demonstrates robust capabilities in modeling and analyzing time-series data [24]. In this study, the BiLSTM was strategically applied to the skip connections of the U-Net, linking the output of the BiLSTM to the corresponding feature maps of the decoder. This integration enhanced the accuracy of the classification and recognition results of the THz time-domain signals.

By leveraging the distinctive features of THz waveforms, the positions of characteristic peaks and troughs were predicted and labeled from two directions. To quickly and precisely annotate THz waveform peaks, a BiLSTM network structure was employed. This BiLSTM structure combined forward and backward LSTMs, enabling predictions and annotations for the positions of signal feature peaks and troughs from both directions [25]. This approach effectively introduced “future” data information for the current time period, enhancing the model’s ability to capture dependencies among temporal features. The BiLSTM network structure is illustrated in Figure 5.

In Figure 5, the LSTM layer incorporates three key gates: the forget, input, and output gates [26]. The forget gate is represented by $f_{t}$; the input gate is represented by $i_{t}$; and the output gate is represented by $o_{t}$. The forget gate determines the extent to which information is discarded from the cell state $c_{t-1}$, with each value ranging between 0 and 1: 0 signifies complete forgetting, while 1 indicates no change. The input gate controls how much information about the current state is input into the memory cell, and the output gate regulates the extent of output based on the current cell state $c_{t-1}$. The specific implementation formulas are presented as follows:(1)$$f_{t}=\sigma \left(w_{f}x_{t}+u_{f}h_{t-1}+b_{f}\right),$$
(2)$$i_{t}=\sigma \left(w_{i}x_{t}+u_{i}h_{t-1}+b_{i}\right),$$
(3)$$o_{t}=\sigma \left(w_{o}x_{t}+u_{o}h_{t-1}+b_{o}\right),$$
(4)$$c_{t}=f_{t}\cdot c_{t-1}+i_{t}\cdot tanh\left(w_{c}x_{t}+u_{c}h_{t-1}+b_{c}\right),$$
(5)$$h_{t}=o_{t}\cdot tanh\left(c_{t}\right),$$
where $x_{t}$ is the input vector; $h_{t}$ is the hidden layer state; $c_{t}$ is the cell state; σ is the sigmoid function; $w_{f}$, $w_{i}$, $w_{o}$, and $w_{c}$ represent the input weights; $u_{f}$ $u_{i}$, $u_{o}$, and $u_{c}$ represent the recurrent weights; and $b_{f}$*,* $b_{i}$, $b_{o}$, and $b_{c}$ represent the bias weights.

The symmetrical structure of U-Net enhances its ability to preserve signal features. It primarily consists of an encoder, a decoder, and skip connections [27,28]. The THz data of different defect types are input into the model to obtain the probability distribution of the defects at different positions. The THz data input undergoes four downsampling stages and four upsampling stages. The downsampling stages include a series of convolutional networks with 1 × 5 convolutions. The convolution layers employ the same padding to maintain feature resolution before and after convolution. Each convolution is followed by a normalization layer and a rectified linear unit (ReLU) layer. After each convolution, downsampling is performed using a 1 × 2 max-pooling layer with a stride of 2. In each downsampling step, the network doubles the number of feature maps and halves their channel dimension. The upsampling stages use transpose convolutions for feature map upsampling. The transpose convolution has a size of 1 × 5, a stride of 2, and uses the same padding. The transpose convolution is followed by a convolution layer, a normalization layer, and an ReLU layer. In each upsampling step, the number of feature maps is halved, and their channel dimension is doubled. The upsampling feature and the downsampling feature after BiLSTM are combined for a subsequent convolution. The final output values are transformed into a probability distribution in the range 0–1, with a sum of 1, using the Softmax function. The Softmax layer is intricately linked to the fully connected layer to determine the probabilities in the final layer of the model. The architecture of the U-Net-BiLSTM network model is depicted in Figure 6.

## 3. Results and Discussion

In this study, five neural network models were trained, validated, and tested, with all experiments being conducted on a computer equipped (Lenovo, Beijing, China) with Intel^®^ Core^TM^ i7-10700KF CPU @ 3.80 GH, 128 GB RAM, NVIDIA GeForce RTX 3080 Ti GPU. The five neural network models utilized the same dataset as the input.

### 3.1. Evaluation Metrics

Four statistical metrics were chosen to assess the results of neural network recognition as follows: accuracy (***Acc***), sensitivity (***Sen***), specificity (***Spe***), precision (***Pre***), and the comprehensive evaluation metric, ***F1***. These four metrics were computed using the following fundamental classification metrics: namely, true positive (***TP***), true negative (***TN***), false positive (***FP***), and false negative (***FN***) [29]. ***TP*** indicates that the prediction is positive and that the actual result is also positive. ***TN*** indicates that the prediction is negative, and the result is actually negative. ***FP*** indicates that the prediction is positive, and the actual result is negative. ***FN*** indicates that the prediction is negative, and the actual result is positive. The formulas are as follows: (6)$$Acc=\frac{TP+TN}{TP+TN+FP+FN},$$
(7)$$Sen=\frac{TP}{TP+FN},$$
(8)$$Spe=\frac{TN}{TN+FP},$$
(9)$$Pre=\frac{TP}{TP+FP},$$
(10)$$F_{1}=\frac{2\times Pre\times Sen}{Pre+Sen},$$
where ***Acc*** is the proportion of correctly classified positive and negative samples among all samples; ***Sen*** is the proportion of correctly classified positive samples among all positive samples; ***Spe*** is the proportion of correctly classified negative samples among all negative samples; ***Pre*** is the proportion of correctly classified positive samples among all predicted positive samples; and the ***F1*** score comprehensively evaluates ***Sen*** and ***Pre***. The results, presented in Table 2, indicate that the U-Net-BiLSTM network exhibits a superior recognition performance, followed by the U-Net, BiLSTM, CNN, and ResNet networks.

In order to further evaluate the recognition ability of the U-Net-BiLSTM network towards different defects of a multi-layer thermal protection structure, the network defect recognition loss value and accuracy of the U-Net network connecting different numbers of BiLSTM were counted. According to the U-Net-BiLSTM network structure in Figure 6, the number of BiLSTM is reduced from top to bottom. The results are shown in Figure 7.

It can be seen from Figure 7 that, as the number of iterations increases, the loss function of the U-Net-BiLSTM network with four BiLSTMs decreases the fastest, and the accuracy function is also located above that of other networks, ranking second. Therefore, the BiLSTM layer is added to each jump connection layer of the U-Net-BiLSTM network built in this paper.

### 3.2. Recognition Results

This study adjusted the fabricated samples to simulate defects of an unknown thickness. Utilizing a THz-TDS system with a detection step of 1 mm, the TDS signals collected from defect-free positions, positions with delamination defects, and positions with debonding defects were employed as the input vectors for training, validating, and testing the three network models. During validation, the trained model’s output for each signal was categorized into one of the following areas: defect-free, delamination defect, and debonding defect. The neural networks’ output for the three types of defects is shown in Figure 8.

As shown in Figure 8, the defect thickness d can be computed by assessing the time delay of characteristic peaks and troughs at the defect position [30], calculated as follows:(11)$$d=\frac{c\times \Delta t}{2n},$$
where c is the speed of light in a vacuum; Δt is the time delay of characteristic peaks and troughs at the defect position; and n is the refractive index of the material in the THz range. Given that the defect position involved an air gap, the refractive index was set to 1.

Based on the output of all point-wise signals, the three-dimensional reconstructed images depicting defect thickness are presented in Figure 9. The purple is the upper surface of the multi-layer lightweight composite structures, the blue is the defect, and the red is the metal substrate.

Using Equation (11), the average thickness of the five defect positions in the delamination and debonding defect samples was computed. Given a detection step of 1 mm, each pixel’s area was 1 mm^2^, which was used to calculate the defect area at the five positions. The calculated defect thicknesses and areas are presented in Table 3, where the thickness error represents the maximum error among the five defect positions, and the area error is the sum of the error pixel points among the five defect positions. By analyzing the three-dimensional reconstructed defect images in Figure 9 and the thickness and area identification results in Table 3, it can be inferred that the U-Net-BiLSTM network’s results are the closest to those deriving from manual identification. The variation in the defect thickness identification error between the U-Net-BiLSTM network and manual identification is less than 0.06 mm. For the delamination defect sample, the defect area error is 57 pixels compared to manual identification, and, for the debonding defect sample, the defect area error is 73 pixels compared to manual identification, both of which are smaller than the identification errors of the CNN, ResNet, U-Net, and BiLSTM networks.

## 4. Conclusions

In this paper, the U-Net-BiLSTM network is used to identify the defects of THz detection waveforms of multi-layer lightweight composite structures. According to the relationship between the THz waveform’s flight time and the defects’ thickness, defect depth is calculated. According to the pixel points of the identified defects, the defect area is calculated, and the three-dimensional defect map is constructed according to the recognition results. The U-Net-BiLSTM network integrated BiLSTM into the skip connections of U-Net, connecting the BiLSTM output to the corresponding feature maps of the decoder. This approach yielded more precise results for defect classification and thickness recognition in THz time-domain signals. The accuracy of the U-Net-BiLSTM network is 99.45%, and the F1 score is 99.43%, demonstrating superior performance when compared with the CNN, ResNet, U-Net, and BiLSTM networks. Furthermore, three-dimensional defect reconstructions based on network recognition results offered a more intuitive visualization. The U-Net-BiLSTM network results to be closely aligned with manual identification, featuring defect thickness identification errors of 0.0538 and 0.0574 mm for the delaminated and debonding defect samples, respectively. The defect area errors were 57 and 73 pixels for the delamination and debonding defects, respectively, compared to manual identification.

The three-dimensional diagram drawn based on the U-Net-BiLSTM network structure proposed in this study shows excellent intelligent recognition of defect areas. Because the U-Net-BiLSTM network is composed of U-Net and BiLSTM, it may lead to no single network with a high efficiency but to one with a high recognition accuracy. In the future, the establishment of a defect feature library without manual judgment of the reflected pulse would provide new ideas for improving the quantitative detection of composite material defects using THz technology.

## Figures and Tables

**Figure 1 materials-17-00839-f001:**
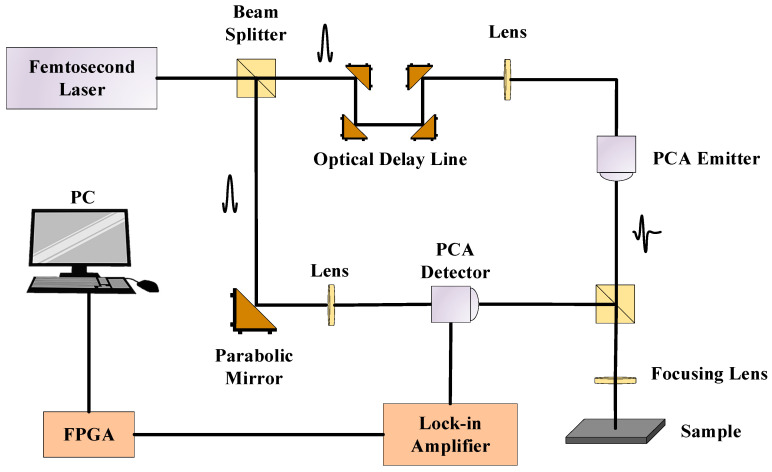
THz-TDS system.

**Figure 2 materials-17-00839-f002:**
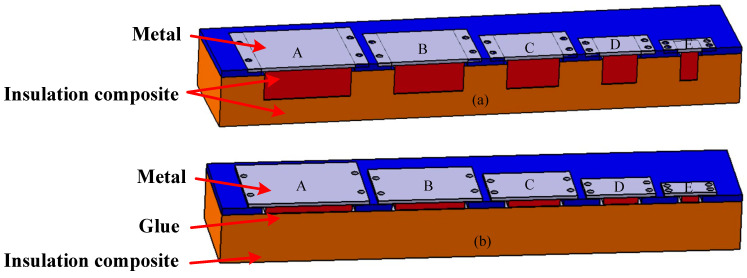
Conceptual design of the multi-layer lightweight composite structure samples. (**a**) Delamination defect; and (**b**) debonding defect.

**Figure 3 materials-17-00839-f003:**
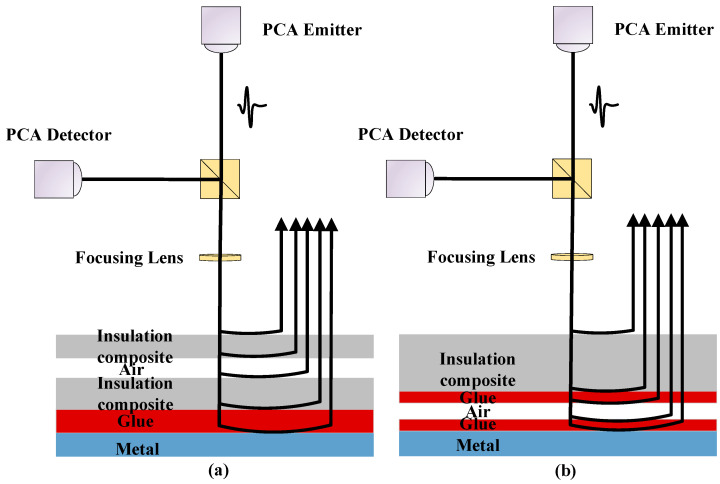
THz wave propagation in multi-layer lightweight composite structure. (**a**) THz wave propagation diagram of a delamination defect sample; (**b**) THz wave propagation diagram of a debonding defect sample.

**Figure 4 materials-17-00839-f004:**
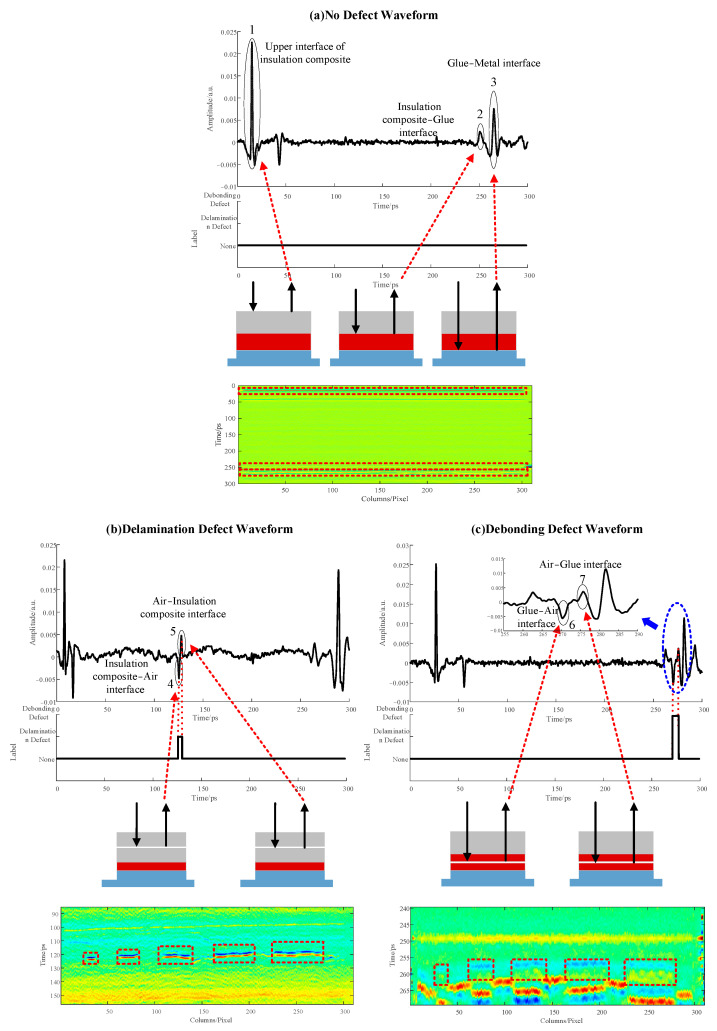
THz reflection waveforms and signal annotation in multi-layer lightweight composite structures with different defect types. (**a**) Terahertz reflection waveform and signal marking of the defect-free sample; (**b**) Terahertz reflection waveform and signal marking of the delamination defect sample; (**c**) Terahertz reflection waveform and signal marking of the debonding defect sample.

**Figure 5 materials-17-00839-f005:**
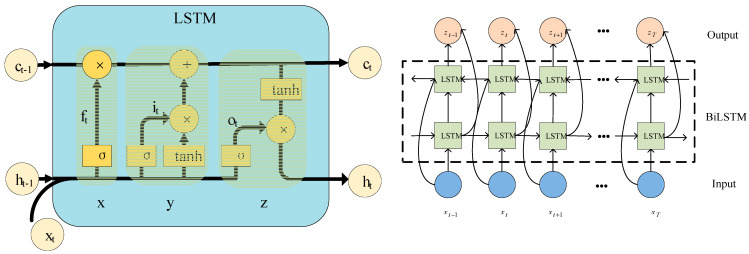
BiLSTM network structure.

**Figure 6 materials-17-00839-f006:**
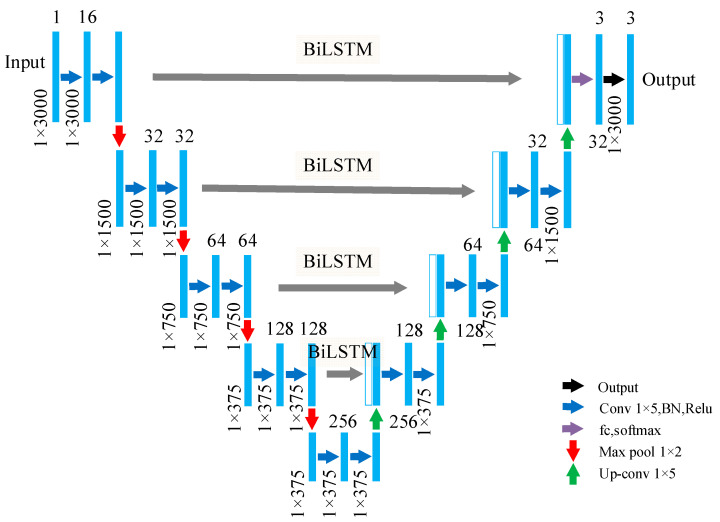
U-Net-BiLSTM network model.

**Figure 7 materials-17-00839-f007:**
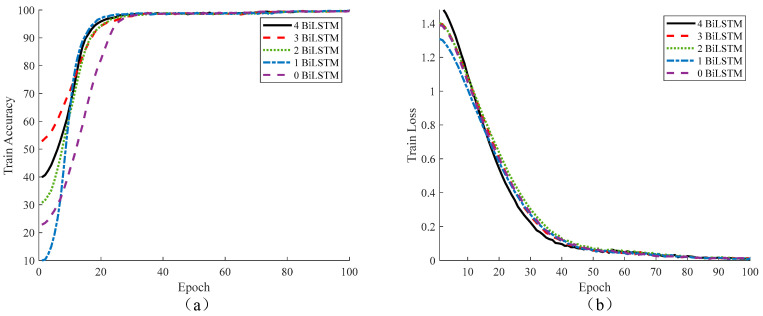
Iterative accuracy and loss value of different network defect recognition training: (**a**) accuracy and (**b**) loss value.

**Figure 8 materials-17-00839-f008:**
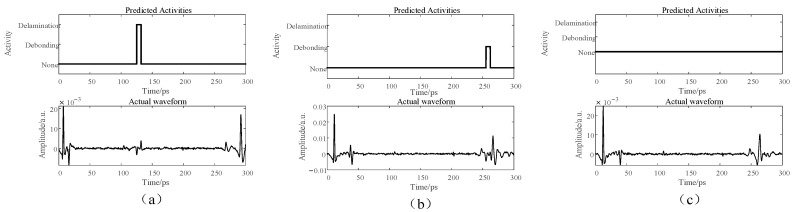
Neutral network recognition results of THz signals from different defects. (**a**) Recognition results for a delamination defect waveform; (**b**) recognition results for a debonding defect waveform; and (**c**) recognition results for a defect-free waveform.

**Figure 9 materials-17-00839-f009:**
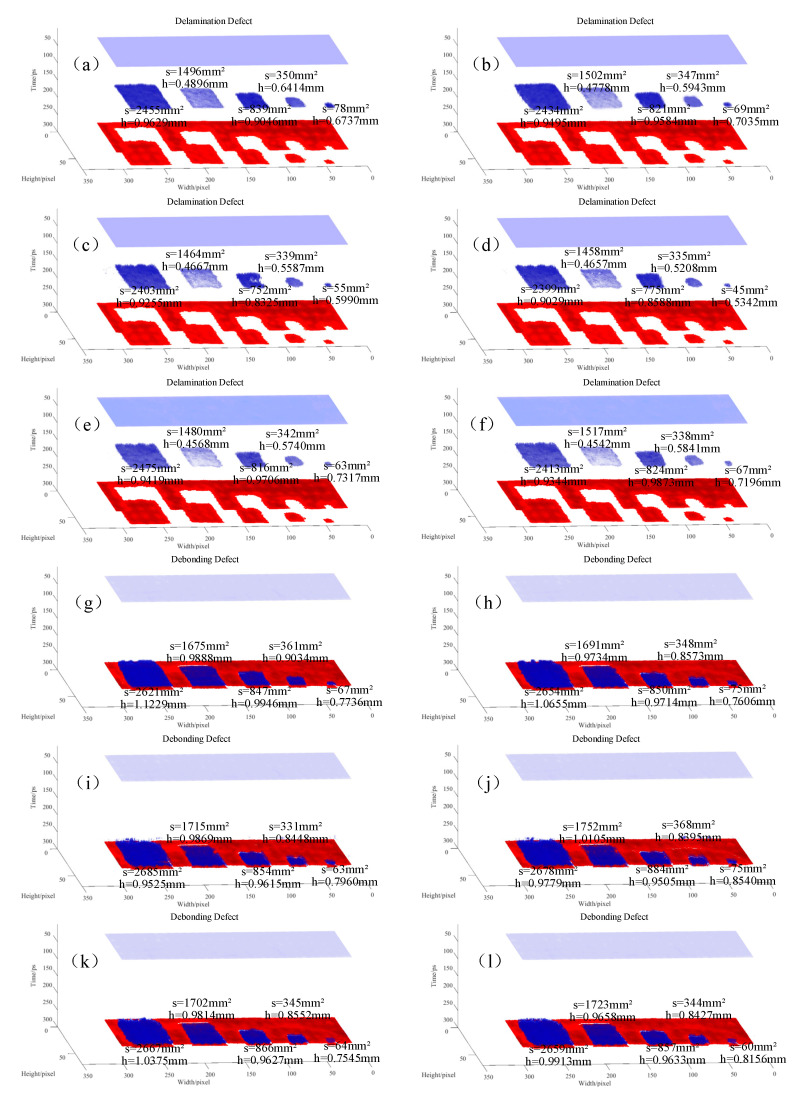
Three-dimensional reconstructed images of neural network recognition results. (**a**–**f**) show the identification results for the delamination defects in a multi-layer lightweight composite structure using manual identification, a U-Net-BiLSTM network, a CNN network, a ResNet network, a U-Net network, and a BiLSTM network, respectively. (**g**–**l**) show the identification results for debonding defects in a multi-layer lightweight composite structure using manual identification, a U-Net-BiLSTM network, a CNN network, a ResNet network, a U-Net network, and a BiLSTM network, respectively.

**Table 1 materials-17-00839-t001:** Predetermined defect specifications for the multi-layer lightweight composite structure samples.

Defect Type	Area A (mm × mm)	Area B (mm × mm)	Area C (mm × mm)	Area D (mm × mm)	Area E (mm × mm)
Debonding	50 × 50	40 × 40	30 × 30	20 × 20	10 × 10
Delamination	50 × 50	40 × 40	30 × 30	20 × 20	10 × 10

**Table 2 materials-17-00839-t002:** Comparison of network recognition evaluation metrics.

Model	Acc (%)	Sen (%)	Spe (%)	Pre (%)	F1 (%)
U-Net-BiLSTM	99.45	99.39	99.51	99.48	99.43
CNN	98.07	97.21	98.93	98.92	98.06
ResNet	97.34	95.49	99.14	99.13	97.27
U-net	98.76	98.87	98.65	98.63	98.75
BiLSTM	98.68	98.47	98.90	98.88	98.67

**Table 3 materials-17-00839-t003:** Recognition results for defect thickness and defect area.

	Method	A	B	C	D	E	Error
Delamination defect thickness (mm)	Artificial	0.9629	0.4896	0.9046	0.6414	0.6737	—
	U-net-BiLSTM	0.9495	0.4778	0.9584	0.5943	0.7035	**0.0538**
	CNN	0.9255	0.4667	0.8325	0.5587	0.5990	0.0827
	ResNet	0.9029	0.4657	0.8588	0.5208	0.5342	0.1395
	U-net	0.9419	0.4568	0.9706	0.5740	0.7317	0.0674
	BiLSTM	0.9344	0.4542	0.9873	0.5841	0.7196	0.0827
Debonding defect thickness (mm)	Artificial	1.1229	0.9888	0.9946	0.9034	0.7736	—
	U-net-BiLSTM	1.0655	0.9734	0.9714	0.8573	0.7606	**0.0574**
	CNN	0.9525	0.9869	0.9615	0.8448	0.7960	0.1704
	ResNet	0.9779	1.0105	0.9505	0.8395	0.8540	0.1450
	U-net	1.0375	0.9814	0.9627	0.8552	0.7545	0.0854
	BiLSTM	0.9913	0.9658	0.9633	0.8427	0.8156	0.1316
Delamination defect area (mm^2^)	Artificial	2455	1496	839	350	78	—
	U-net-BiLSTM	2434	1502	821	347	69	**57**
	CNN	2403	1464	752	339	55	205
	ResNet	2399	1458	775	335	45	206
	U-net	2475	1480	816	342	63	82
	BiLSTM	2413	1517	824	338	67	101
Debonding defect area (mm^2^)	Artificial	2621	1675	847	361	67	—
	U-net-BiLSTM	2654	1691	850	348	75	**73**
	CNN	2685	1715	854	331	63	145
	ResNet	2678	1752	884	368	75	186
	U-net	2667	1702	866	345	64	111
	BiLSTM	2659	1723	857	344	60	120

## Data Availability

The data used to support the findings of this study are available from the corresponding author upon request.

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
