# Peer review of "Quantitative Detection of Defects in Multi-Layer Lightweight Composite Structures Using THz-TDS Based on a U-Net-BiLSTM Network"

_materials, 2024, doi:10.3390/ma17040839_

Round 1

Reviewer 1 Report

Comments and Suggestions for Authors

This manuscript demonstrates detection of defects in samples of multilayer  composite  structures with the THz-TDS instrument and the use of network model that integrates BiLSTM and U-Net structure for  better identification of types of defects and thickness recognition.  I found this paper interesting for a broader audience, its results potentially very useful.  I recommend publication of the manuscript in Materials after authors consider my comment below.

Little or nothing is said about experimental setup. Although their previous paper is cited for detailed description, I am suggesting explanations to be added about the instrument used for detecting defects in samples, about PCA emitter and detector, on guiding and focusing THz pulses, how such  fast detection and high time resolution was achieved. How the images of defects spread inside the large sample (50 x 300 mm) at about the same distance from the top surface (Fig. 4) were taken? Besides, instrument in the cited reference is commercial while this one is sad to be home made, “developed in the laboratory”.

Note that in Fig. 4, peaks 6 and 7 are not between peaks 2 and 3 shown on Fig 4a. They are behind/after peak 2.  

Author Response

请参阅附件

Reviewer 2 Report

Comments and Suggestions for Authors

1.     Extend the literature review, and consider including the following paper to provide readers with insights into recent real-life applications.

2.     In the introduction, the authors should clearly describe their contribution and contrast it with the published work. Add highlights to the manuscript.

3.     Provide a detailed explanation of the novelty and how your work advances the field.

4.     The analysis of the results is weak. Please provide a more comprehensive analysis in the Discussion section.

5.     The manuscript lacks validation for experimental and numerical results against other valid findings in the literature. Compare your results with existing techniques and provide a thorough evaluation.

6.     Enlarge the letters in the images, and improve the overall image quality for better clarity and visual interpretation, especially in Figures 4, 8, and 7.

7.     Improve the quality of Figures 4, 8, and 7.

8.     Compare the performance of the proposed network model design against previously reported model(s).

9.     Discuss future work plans.

10. Improve the language of the paper.

Reviewer 3 Report

Comments and Suggestions for Authors

Reviewer's report on the paper titled “Quantitative Detection of Defects in Multi-Layer Lightweight  Composite Structures Using THz-TDS Based on a U-Net-BiLSTM Network”

The paper “Quantitative Detection of Defects in Multi-Layer Lightweight  Composite Structures Using THz-TDS Based on a U-Net-BiLSTM Network” is an original contribution, which is joined with the scope of MDPI Journal Materials. Three-dimensional THz defect images were successfully reconstructed to provide a quantitative defect detection for multi-layered composite structure by leveraging defect classification and thickness recognition. The accuracy of the proposed U-Net-BiLSTM network was evaluated for the typical defect identification. A model tailored for THz spectroscopy data, were obtained by optimizing the network structure. The following improvements/complications should be done to made the paper more clear for understanding:   

1. The current paper should be completed by the information regarding the multi-layered lightweight composite structural materials, which were considered in the current study. 

2. Abstract of the paper should be expanded by the information regarding multi-layered lightweight composite materials were considered. 

3. Chapter 1. Introduction should be expanded by the clear definition of goal of the current investigation and its basis. Clear formulations of the tasks should be solved to obtain the above mentioned goal should be clearly formulated also. 

4. All the values included at the the equations 1) – 11), should be described in the designations below or in the text. 

5. Figure 2 should be supplied by the additional information regarding the specimens are shown. Cross-sections and geometric parameters of the considered specimens should be added. 

6. Chapter 4. Conclusions should be rewritten as a number of statements, based on the results were obtained.  Now it looks like more as an abstract of the paper. Some information regarding the probable limitation of the proposed model depending on the properties of the considered composite materials should be added. 

7. The list of references should be significantly expanded. 

Round 2

Reviewer 2 Report

Comments and Suggestions for Authors

Congratulations on the paper and the effort put into responding to reviews.

Reviewer 3 Report

Comments and Suggestions for Authors

Repeated reviewer's report on the paper titled “Quantitative Detection of Defects in Multi-Layer Lightweight  Composite Structures Using THz-TDS Based on a U-Net-BiLSTM Network”

The paper “Quantitative Detection of Defects in Multi-Layer Lightweight  Composite Structures Using THz-TDS Based on a U-Net-BiLSTM Network” was significantly improved after the first review. All the comments were addressed.